# A ketone/alcohol polymer for cycle of electrolytic hydrogen-fixing with water and releasing under mild conditions

Ryo Kato[1], Keisuke Yoshimasa[1], Tatsuya Egashira[1], Takahiro Oya[1], Kenichi Oyaizu[1] & Hiroyuki Nishide[1]

Finding a safe and efficient carrier of hydrogen is a major challenge. Recently, hydrogenated organic compounds have been studied as hydrogen storage materials because of their ability to stably and reversibly store hydrogen by forming chemical bonds; however, these compounds often suffer from safety issues and are usually hydrogenated with hydrogen at high pressure and/or temperature. Here we present a ketone (fluorenone) polymer that can be moulded as a plastic sheet and fixes hydrogen via a simple electrolytic hydrogenation at $-1.5\,V$ (versus Ag/AgCl) in water at room temperature. The hydrogenated alcohol derivative (the fluorenol polymer) reversibly releases hydrogen by heating ($80\,°C$) in the presence of an aqueous iridium catalyst. Both the use of a ketone polymer and the efficient hydrogen fixing with water as a proton source are completely different from other (de)hydrogenated compounds and hydrogenation processes. The easy handling and mouldable polymers could suggest a pocketable hydrogen carrier.

[1] Department of Applied Chemistry, Waseda University, 3-4-1 Okubo, Shinjuku, Tokyo 169-8555, Japan. Correspondence and requests for materials should be addressed to H.N. (email: nishide@waseda.jp).

Hydrogen is produced at chemical facilities and from a great variety of potential sources, subsequently stored and then delivered to an end-use application where water is the sole combustion or reaction product[1–3]. Conventional storing and carrying methods of hydrogen use high pressure or cryogenic tanks; however, these methods intensively consume energy and are accompanied by inherent safety risks, such as explosions (for example, by collisions), permeability of hydrogen and embrittlement of the tank walls[4]. Thus, hydrogen-storing and/or -carrying material systems with high safety, ease of handling and low energy loss during the hydrogen storing/carrying/releasing are highly demanded.

Porous materials such as carbon powders[5,6] and metal organic frameworks[7,8] adsorb hydrogen. However, the adsorption has been attributed to weak interactions of hydrogen with the surface of porous solids, of which operation was often restricted at very low temperatures and/or at very high pressures[9]. Recently, organic compounds with reversible hydrogen-storing capability have been intensively studied; the typical examples are nitrogen heterocycles[10,11], formaldehyde water[12] and amide compounds[13]. These materials feature a high stability of the corresponding hydrogenated compounds due to the chemical bond formation and high hydrogen storage capacity of 2–6 wt% in the exemplified compounds. However, these compounds are mostly liquids or used as solutions and often suffer from safety issues of toxicity, flammability and volatility. Furthermore, their hydrogenation mostly proceeds at high pressure of hydrogen via highly energy-consuming processes. Although some of these liquid organic compounds have been tested as hydrogen storage materials on an industrial scale to be used in the existing infrastructure for oil storage and transportation, organic liquids are not always suitable to be examined in small-scale applications as hydrogen storage materials. They need vessels or sealed tanks operated at high pressure and/or temperature and often encounter difficulty in their separation from the evolved hydrogen gas. A pocketable and safe hydrogen carrier system, just similar to rechargeable batteries as a portable energy source, is anticipated.

Alcohols have been studied as one-way hydrogen-supplying molecules[14,15], where iridium complexes were often used as effective catalysts for the hydrogen evolution under relatively mild conditions[16,17]. In this study we focus on fluorenol, which is an alcohol derivative of fluorene and holds two hydrogen atoms with C–H and O–H chemical bonds. We note that fluorenol efficiently releases hydrogen with the iridium catalyst under mild conditions to yield fluorenone, the corresponding ketone form of fluorenol. We have found that fluorenone turned to the dianion species by applying − 1.5 V (versus Ag/AgCl) and was protonated with water to return to the original fluorenol. We have further extended the hydrogenation/dehydrogenation cycle of the fluorenone/fluorenol couple to the polymeric analogue, for example, a bendable polymer sheet, and examined electrolytic hydrogenation with water and the following hydrogen release from the polymer sheet by simply heating it in the presence of an aqueous catalyst. The fluorenone/fluorenol polymer was repeatedly hydrogenated and dehydrogenated with cyclability. Hydrogen is reversibly fixed in and released from the organic polymer (the bendable sheet) under mild conditions: easy handling, mouldability, robustness, non-flammability and low toxicity are inherent advantages of the polymers.

## Results

**One-pot preparation of the fluorenone and fluorenol polymer.** The fluorenone and fluorenol polymers were moulded via a cross-linking reaction of a bifunctional fluorenone or fluorenol derivative, 2,7-bis[2-(diethylamino)ethoxy]-9-fluorenone or -fluorenol, and a three-functional cross-linker, 1,3,5-tris(bromomethyl)benzene (Fig. 1c; equimolar feed ratio of the

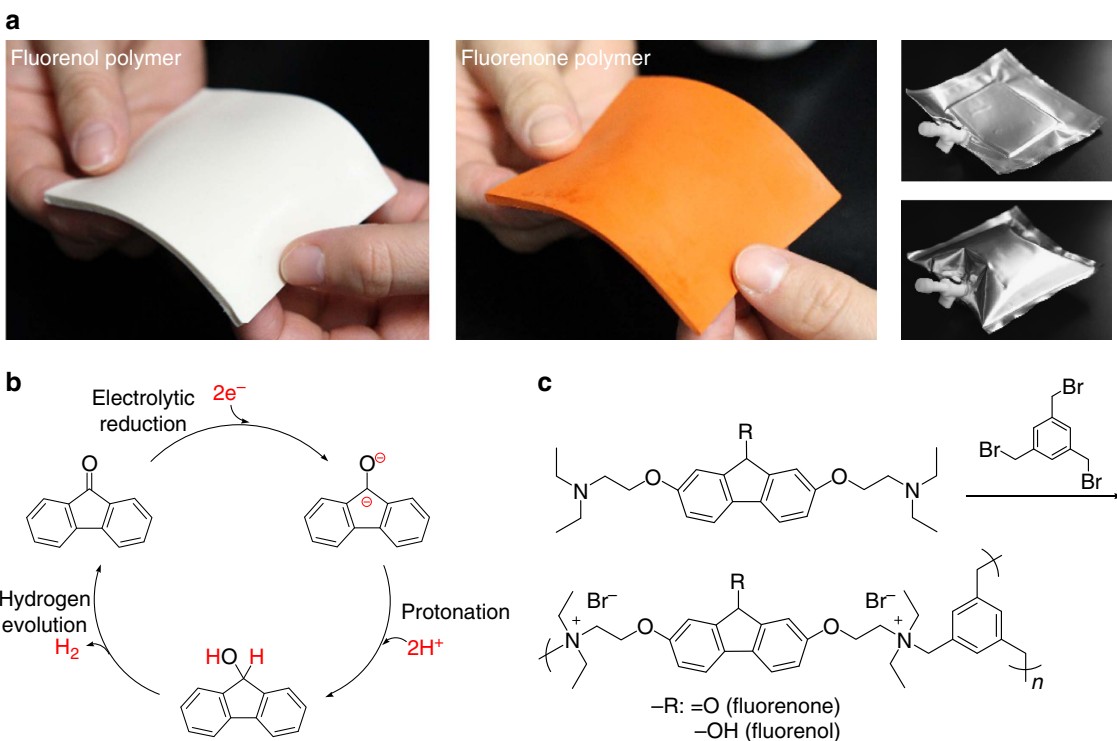

**Figure 1 | Hydrogen fixing and releasing by the fluorenone/fluorenol polymer.** (**a**) A sheet of the fluorenone and fluorenol hydrogel on a 5 g scale and the fluorenol sheet sealed up with a gas-barrier bag (after hydrogen releasing). (**b**) Schematic representation of hydrogen-fixing and -releasing cycle. (**c**) Preparation scheme of the fluorenone and fluorenol polymer.

functional groups), in a Teflon boat by simply heating (90 °C) their *N*-methylpyrrolidone solution and washing it with *N*-methylpyrrolidone, water and methyl alcohol (Supplementary Methods). A typical yield of the insoluble network or gel part (gel fraction) was 86% (the median of five specimens). The chemical structures of fluorenone, fluorenol and quaternized ammonium moieties represented in Fig. 1c were supported by [13]C cross-polarization magic angle spinning–nuclear magnetic resonance (CP/MAS NMR) spectroscopy (Supplementary Fig. 1). A network structure of the polymer was analysed by dynamic mechanical measurement (Supplementary Fig. 2). The storage modulus was much larger than the loss modulus, suggesting a mechanically tough gel formation with adequacy for a pocketable or a facile transportable hydrogen carrier (for the moulded example of a bendable sheet via the one-pot reaction, see Fig. 1a,b). Standing experiments of the fluorenone and fluorenol polymers in an air atmosphere at 80 °C for 30 days did not show any degradation or any chemical structure change detected in [13]C CP/MAS NMR spectroscopy (and the amount of evolved hydrogen gas from the fluorenol polymer was not significantly decreased after the standing), to support durability of the polymers as a hydrogen carrier. Neither the fluorenone nor fluorenol polymers were soluble in water, but were hydrophilic and swollen; they uptook water with 25–35 wt% on the air of a humidity of 30% and with 50–60 wt% in water to be the hydrogels.

**Electrolytic hydrogenation or hydrogen fixation**. We have previously reported that the polymers of aromatic ketone derivatives, such as anthraquinone polymers, are reversibly reduced and applicable as an anode-active material in organic-based rechargeable devices[18–21]. Fluorenone is an aromatic ketone compound and a two-step reversible wave in the cyclic voltammogram was preliminarily reported in water-free solvents[22], suggesting two negative charge storage per fluorenone unit. We have found, in this study, that fluorenone turns back to fluorenol through the very facile electrolytic two-electron reduction and two-proton addition in water.

*In situ* preparation of the fluorenone polymer on a glassy carbon substrate gave an electrode homogeneously coated with the fluorenone polymer with *ca.* 1 μm layer thickness. The fluorenone polymer layer was swollen but did not elute out in water, acetonitrile (AN), or their mixtures. In the AN electrolyte (0.1 M $(C_4H_9)_4NPF_6$), the fluorenone polymer electrode exhibited two quasi-reversible redox waves at −1.3 and −1.7 V (versus Ag/AgCl) in the cyclic voltammetry (CV; Supplementary Fig. 3a). The electrolytic reduction capacity of the fluorenone polymer layer reached 67 mA h g$^{-1}$, which was 92% of the calculated capacity for the two-electron reduction: almost all fluorenone units stored two negative charges throughout the whole polymer layer (Fig. 2). This charging proceeded quantitatively, even at a rapid charging rate of 10 C (or charging for 6 min), indicating that charge propagated efficiently throughout the polymer layer based on the redox gradient-driven and the rapid charge self-exchange reaction among the fluorenone units (we have previously described a similar charge propagation and storage for redox-active polymers[23–25]).

The electrochemical study of the fluorenone polymer layer was then carried out in protic electrolyte solutions. The addition of a drop of water to the AN electrolyte immediately caused, in the CV, disappearance of the oxidation peaks at negative potential (Supplementary Fig. 3b). The fluorenone polymer layer was monitored with infrared by applying −1.5 V in AN/water (vol 5/1) electrolyte: The absorption ascribed to $v_{C=O}$ at 1,710 cm$^{-1}$ decreased as the passed charge increased (Fig. 2, inset), indicating the decrease in fluorenone content in

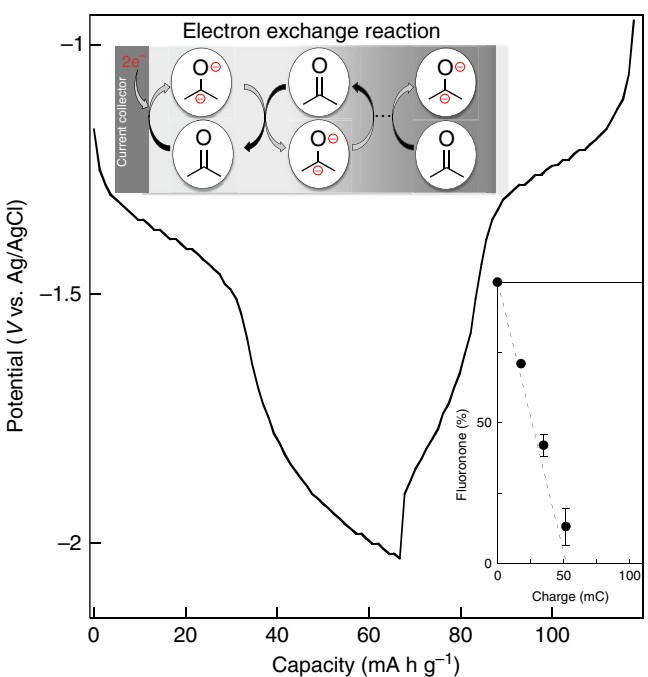

**Figure 2 | Charge storage and electrolytic hydrogenation of the fluorenone polymer.** Charging/discharging curves of the fluorenone polymer at a rate of 10 C measured in the AN electrolyte. Inset: schematic image of charge propagation in the polymer layer and conversion plots of the fluorenone to fluorenol unit in the fluorenone polymer layer electrolytically reduced in the AN/water electrolyte. Theoretical capacity of the fluorenone polymer-coated electrode was 52 mC per 0.2 mg polymer. The dashed line indicates the theoretical conversion with the passed charge; the error bars for the s.d. calculated from five samples.

the polymer through the electrolytic reduction, or protonation or hydrogenation. The coulombic efficiency was, for example, 87% after the passage of equivalent charge, estimated with the conversion of the fluorenone unit.

Bulk reduction of the monomeric fluorenone dissolved in the AN/water (vol 5/1) electrolyte was examined by using a glassy carbon working electrode and applying −1.5 V. The electrolytic reduction progressed almost quantitatively (coulombic efficiency 97%). The product was isolated (yield 95%) and identified to be fluorenol by [1]H NMR spectroscopy (Supplementary Fig. 4). The same reduction of fluorenone was carried out by using the AN/$D_2O$ solution, to almost quantitatively yield two-deuterated fluorenol $C_{13}OH_8D_2$ ($m/z =$ calcd for 184.23). The deuterated fluorenol was heated at 80 °C with the aqueous iridium catalyst (described in the succeeding section). The retention time of the gas evolved from the deuterated fluorenol was identical to that of $D_2$ in gas chromatography, clearly supporting the electrolytic deuteration (hydrogenation) of fluorenone with $D_2O$ (water) and deuterium (hydrogen) evolution from the fluorenol.

To discuss the rapid and quantitative hydrogenation reaction in the fluorenone polymer layer, we investigated the [1]H NMR of the mixture of fluorenol and the fluorenone dianion (the latter was prepared by the electrolytic reduction of fluorenone) (Supplementary Methods). The peaks of hydroxyl (H$^a$) and cyclopentane (H$^b$) protons of fluorenol were broadened and shifted downfield by increasing the dianion content (Fig. 3a), indicating a proton-exchange reaction between the fluorenol and the fluorenone dianion. The rate constants of the exchange were estimated with the relaxation time (*T*) (Supplementary Fig. 5 and Table 1). These proton-exchanging rate constants are in the order of $10^3$–$10^4$ M$^{-1}$ s$^{-1}$ for fluorenol/fluorenone dianion and were

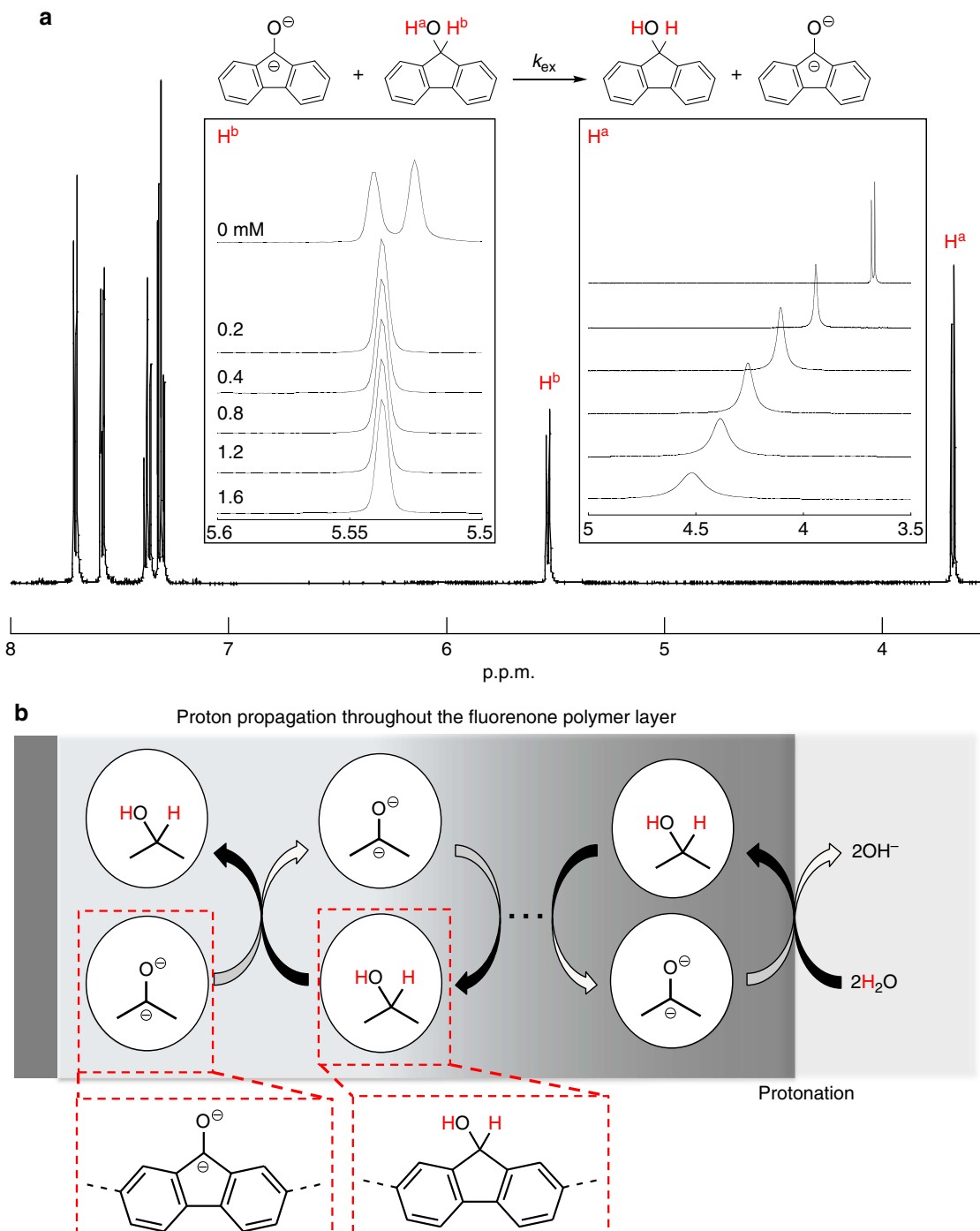

**Figure 3 | Proton exchange reaction of fluorenol/fluorenone dianion.** (**a**) $^1$H NMR measurements of fluorenol in the presence of the fluorenone dianion were conducted in AN-$d_3$. Inset, expanded cyclopentane and hydroxyl proton spectra of fluorenol. (**b**) Schematic image of proton propagation in the polymer layer.

large enough to explain the proton exchange in the fluorenone polymer layer.

Similar proton propagation had been examined using hydroquinone polymers[26,27]; however, most of those were electrochemically inactive in aqueous media because of the very slow electron/proton hopping and of hydrophobic property of the polymers, and the redox active examples have been restricted only to the very thin quinone layer coated on a glassy carbon[27]. The fluorenone polymer hydrogel caused both an electrochemically effective charge and proton propagation or the hoppings (Fig. 2,

inset and Fig. 3b), probably due to an appropriate network structure of the fluorenone units and the hydrophilic gel state. It should be noted here that the electrolytic reduction and the successive hydrogenation of the fluorenone polymer progressed in the presence of aqueous electrolyte at room temperature.

**Hydrogen evolution from the fluorenol polymer.** The fluorenol polymer (10 g) was soaked in an aqueous solution (15 ml) containing the iridium catalyst (157 mg of aqua (6,6'-dihydroxy-2, 2'-bipyridine)(pentamethylcyclopentadienyl)iridium(III) bis

**Table 1 | Hydrogen evolution rate and yield from alcohols.**

| Entry | Alcohol | Solvent (alcohol concn, M) | Initial hydrogen evolution (mL)* | Conversion (%)[†] | Conversion (%)[‡] |
|---|---|---|---|---|---|
| 1[§] | | $H_2O$ (0.90)[¶] | 86 | 36 | 94[‖] |
| 2[#] | | $H_2O$ (saturated) | 2.9 | 1.2 | 2.6 |
| 3** | | $H_2O$ (0.90)[¶] | 76 | 32 | 90 |
| 4 | | $H_2O/t\text{-}C_4H_9OH$ (3/7) (0.01) | 67 | 28 | 82 |
| 5 | | $H_2O/t\text{-}C_4H_9OH$ (3/7) (0.01) | 44 | 18 | 76 |
| 6 | | $H_2O/t\text{-}C_4H_9OH$ (3/7) (0.01) | 31 | 13 | 63 |
| 7 | | $H_2O/t\text{-}C_4H_9OH$ (3/7) (0.01) | 17 | 6.9 | 35 |

*Hydrogen evolution volume (ml) from 10 mmol of the fluorenol or alcohol derivative for the initial 1 h.
†Conversion of alcohols to ketons after 1 h was determined by gas chromatography.
‡The conversion after 5 h.
§The polymer was swollen with the 60 wt% aqueous solution of the catalyst, which corresponded to 0.9 M in concentration of the fluorenol unit and was several thousand times higher than that of the aqueous solution saturated with fluorenol (ca. 0.2 mM).
¶Alcohol unit concentration was calculated by the molar mass of the repeating unit.
‖The conversion was on the average of five reactions with an s.d. of 5.9%.
#The aqueous solution was saturated with fluorenol (0.003 mmol) in water (15 ml) and involved the iridium catalyst (1.4 mol% versus fluorenol).
**The fluorenone/carbon composite (14.3 g) was prepared via hydrogenation with electrolytic reduction and protonation in the aqueous electrolyte.

(triflate)[28]) and heated at 80 °C. Rapid gas evolution from the polymer was ascribed to the elimination of $H_2$ by gas chromatography analysis, which amounted to 309 ml (94% of the formula weight-based mobile hydrogen amount) after 5 h (Table 1 and Supplementary Fig. 6). The fluorenol polymer specimen (5 g) containing the aqueous iridium catalyst was sealed up with a gas-barrier bag and was heated (photo in Fig. 1); the evolved pure hydrogen gas was also analysed by gas chromatography.

Hydrogen evolution rates from the polymer are listed in Table 1, along with those from the monomeric alcohols in the presence of the same iridium catalyst at 80 °C. The rate from the fluorenol polymer was 30 times larger than that from monomeric fluorenol (entries 1 and 2), which is explained by the highly populated fluorenol units in the polymer and by the low solubility of fluorenol in the aqueous solution (footnotes of Table 1).

Among the alcoholic derivatives as hydrogen donors, fluorenol gave the higher rate of hydrogen evolution in comparison with those of other secondary alcohols (entries 4–7), although fluorenol is an aromatic alcohol. X-ray crystallographic structures of fluorenol and fluorenone (Supplementary Fig. 7) depicted that the hydroxyl group and hydrogen atom bound to cyclopentane carbon are located out of the planar (slightly bent) fluorene skeleton, which is in contrast to the totally planar fluorenone structure. This suggests that the elimination of two hydrogens of the secondary alcohol fluorenol to form the π-conjugated planar fluorenone could be thermodynamically favourable.

After the hydrogen evolution, the colourless fluorenol polymer turned a reddish colour (Supplementary Fig. 8a,b). Infrared spectra supported that almost all fluorenol units in the polymer turned into fluorenone units through the dehydrogenation, to yield the fluorenone polymer (characterization in Supplementary Fig. 8c).

**Cycle of electrolytic hydrogenation and hydrogen evolution.** A composite sheet (14.3 g) of the fluorenol polymer with carbon nanofibre as a conductive additive was prepared, to perform the cycling of hydrogen fixing and releasing (Supplementary Methods). The composite sheet was soaked in the AN/water electrolyte and − 1.5 V was applied till a charge of 390 C per g polymer was passed. The sheet was transferred into 15 ml of water containing the iridium catalyst and heated at 80 °C for 5 h. The hydrogen evolution was almost similar to that of the neat fluorenol polymer and the evolved hydrogen (296 ml) almost reached the theoretical hydrogen density held in the composite sheet (entry 3 in Table 1). The composite sheet was then hydrogenated again and this cycle was performed 50 times without any significant deterioration in the hydrogen evolution capacity (Supplementary Fig. 9).

**Discussion**
A hydrogen carrier polymer was prepared using molecular design of fluorenone/fluorenol as a hydrogen-fixing and -releasing unit.

The hydrogen-carrying polymer has inherent advantages, such as mouldability, non-flammability and low toxicity, and only uses water as proton source. These advantages of the polymer lead to an easy handling and portable hydrogen-carrying material for use in, for example, a home/on-site or pocketable hydrogen-supply system. A hydrogen-carrying and -releasing cartridge plate and a hydrogen permselective film could be potential forms for the system.

The electrolytic reduction and protonation with water were successfully combined for the hydrogenation of the polymer, which is largely different from a water splitting; the hydrogenation of the polymer could fix hydrogen through the formation of chemical bonds, whereas a water-splitting process only releases hydrogen. Direct electrolytic hydrogenation with water could be a potential process to reduce energy consumption for the hydrogen-fixing and to eliminate any hydrogen production processes.

The theoretical hydrogen storage capacity of fluorenol is 1.1 wt% (0.29 wt% as the fluorenol polymer system); however, we anticipated its improvement by the use of, for example, piperazine tetranol (2.8 wt%) as the hydrogen-fixing unit.

## Methods

**Reagents and instruments.** Fluorenone, fluorenol and 2,7-bis[2-(diethylamino)ethoxy]-9-fluorenone dihydrochloride were purchased from Tokyo Chemical Industry Co. Aqua (6,6′-dihydroxy-2,2′-bipyridine)(pentamethylcyclopentadienyl)iridium(III)bis(triflate) was purchased from Kanto Chemical Co. $^1H$ NMR spectra were recorded on a JEOL ECX-500 spectrometer with chemical shifts downfield from tetramethylsilane as the internal standard, $^{13}C$ CP/MAS NMR spectra on a JEOL ECX-600 spectrometer, infrared spectra on a JASCO FT/IR-6100 spectrometer, ultraviolet–visible spectra on a JASCO V-550 spectrometer and X-ray crystallographic data on a R-AXIS RAPID. Amount of evolved gas was determined using a gas chromatograph (Shimadzu GC-8AIT, Ar carrier) equipped with a 2 m-long packed column of Molecular Sieve 5A and a recorder (Shimadzu C-R8A Chromatopac Data Processor). For $H_2/D_2$ detection, the gas samples were analysed using a gas chromatograph (He carrier) equipped with a 3 m-long packed column of Shinwa OGO-SP, which was immersed in liquid $N_2$ during the gas chromatograph measurements, and a thermal conductivity detector.

**Mechanical testing of the polymer.** The dynamic mechanical tests of a bar sample of 10 mm width and 40 mm length were performed using a Dynamic Mechanical Analyzer Q-800. Storage modulus, loss modulus and tangent delta were analysed at a frequency of 1 Hz with a ramp speed of $2\,°C\,min^{-1}$.

**Electrochemical measurements.** CV and chronopotentiograms were carried out with a fluorenone polymer-coated glassy carbon plate or disk electrode, a platinum wire and an Ag/AgCl electrode as the working, the counter and the reference electrode, respectively, using an ALS700 electrochemical analyser. The formula weight-based theoretical capacity (in $mA\,h\,g^{-1}$) was calculated according to $1,000\,nFM^{-1}\,(3,600)^{-1}$, where $n$, $F$, and $M$ are stoichiometric numbers of electrons (two in this experiment), Faraday constant and molar mass of the repeating unit, respectively. The redox capacity (in $mA\,h\,g^{-1}$) was obtained employing $1,000\,Qm^{-1}\,(3,600)^{-1}$, where $Q$ and $m$ are charges passed in electrolysis and the loading weight of the fluorenone polymer, which was determined by weighing the anode.

**Data availability.** The data that support the findings of this study are available from the corresponding author upon request.

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

## Acknowledgements

This work was partially supported by Grants-in-Aid for Scientific Research (numbers 24225003 and 15K13713) from MEXT, Japan. R.K. acknowledges the Leading Graduate Program in Science and Engineering, Waseda University, from MEXT, Japan. We thank Professor B. Winther-Jensen for his comments and A. Iwawaki for her experimental assistance.

## Author contributions

R.K., K.O. and H.N. conceived the project. R.K., K.Y., T.E. and T.O. carried out the bulk of the experimental work. R.K., K.O. and H.N. wrote the manuscript. H.N. oversaw all the works. All the authors discussed the results and commented on the manuscript.

## Additional information

**Competing financial interests:** The authors declare no competing financial interests.

