## [Peer review file · Nature Communications]

REVIEWERS' COMMENTS:

Reviewer #1 (Remarks to the Author):

There is now sufficient info for publication. The level of importance of the results isn't up to Nat Comm but would be fine for Nature Chemistry. I suggest transfer to that journal.

Reviewer #2 (Remarks to the Author):

The authors improved the article significantly after the last review process. I have only two minor points:

1. The reproducibility should be given for all experimental data.
2. For calculation of the molar volume of hydrogen the van der Waals equation is more precisely than the ideal gas equation. Then, the molar volume at 25{degree sign}C will be 24.48 l/mol. (not 23.87)

After final improvement I suggest a publication in Nature Comm..

Reply to the reviewer's comments on our manuscript

Response to Reviewer #2:

We appreciate your constructive comments as a way to improve the manuscript. We have added the requested information and modified the description along the comments.

(1)

Comment:

The reproducibility should be given for all experimental data.

Reply:

(i) In the revised manuscript, we added the words of “(the median of 5 specimens)” on line 74, page 6.

(ii) On the footnote of Supplementary Figure 9, we added the sentence of “The cycle performance was tested in twice, and each gas volume was within the error limits”.

The following informations on the reproducibility of data had been given in the original manuscript: The uptook water contents were noted in the range of 25–35 wt% and 50–60 wt% on line 87–88, page 7. The rate constant of the proton exchange was given with 2 significant figures in Supplementary Table 1. In Table 1, f) The conversion was on the average of 5 reactions with the standard deviation of 5.9%.

(2)

Comment:

For calculation of the molar volume of hydrogen the van der Waals equation is more precisely than the ideal gas equation. Then, the molar volume at 25{degree sign}C will be 24.48 l/mol. (not 23.87)

Reply:

Thank you for your pointing out. We used 24.48 l/mol as the molar volume at 25 °C calculated from Charles's law. The collected gas volume was re-calculated, according your comment, using the van der Waals equation, but the conversion was remained the same value with 2 significant figures on line 157–158, page 12 as Table 1.